# Hydrogenation of different carbon substrates into light hydrocarbons by ball milling

Linfeng Li[1,3], Olena Vozniuk [1,3] ✉, Zhengwen Cao[1,2], Pit Losch[1], Michael Felderhoff[1] & Ferdi Schüth [1] ✉

The conversion of carbon-based solids, like non-recyclable plastics, biomass, and coal, into small molecules appears attractive from different points of view. However, the strong carbon–carbon bonds in these substances pose a severe obstacle, and thus–if such reactions are possible at all–high temperatures are required[1–5]. The Bergius process for coal conversion to hydrocarbons requires temperatures above 450 °C[6], pyrolysis of different polymers to pyrolysis oil is also typically carried out at similar temperatures[7,8]. We have now discovered that efficient hydrogenation of different solid substrates with the carbon-based backbone to light hydrocarbons can be achieved at room temperature by ball milling. This mechanocatalytic method is surprisingly effective for a broad range of different carbon substrates, including even diamond. The reaction is found to proceed via a radical mechanism, as demonstrated by reactions in the presence of radical scavengers. This finding also adds to the currently limited knowledge in understanding mechanisms of reactions induced by ball milling. The results, guided by the insight into the mechanism, could induce more extended exploration to broaden the application scope and help to address the problem of plastic waste by a mechanocatalytic approach.

Conversion of different solid, carbon-containing substrates, such as plastic waste, lignocellulosic biomass, or coal, to lower molecular mass compounds has been a tremendous challenge for more than a century. If such processes were available, coal and biomass could easily enter the value streams of the chemical industry, and plastic waste could be converted to valuable raw materials instead of ending up in the environment. Coal gasification or liquefaction had already intensely been studied in the early years of the twentieth century, and the same holds for the conversion of wood[5,9–11]. The standard process, which has emerged from these early studies, is steam reforming, which for coal is operated at about 1000 °C and results in the formation of synthesis gas, which needs to be further converted. For coal, the Bergius process, using temperatures of 450–500 °C and hydrogen pressures of several 100 bar, is a viable alternative to steam reforming, resulting directly in the formation of hydrocarbons[6]. For plastic waste, currently pyrolysis is being explored for so-called "chemcycling", which proceeds under an inert atmosphere at temperatures of typically 500 °C[7,8]. Available conversion technologies for the production of low molecular weight compounds from carbon-containing solids thus require high temperatures and harsh conditions[1–4]. It would be very interesting to develop pathways proceeding at low temperatures, preferably avoiding any external heat sources.

Mechanochemistry has been shown in recent years to be a suitable method to initiate unusual reactions or to allow reactions under

[1]Department of Heterogeneous Catalysis, Max-Planck-Institut für Kohlenforschung, Kaiser-Wilhelm-Platz 1, 45470 Mülheim an der Ruhr, Germany. [2]Qingdao Institute of Bioenergy and Bioprocess Technology, Chinese Academy of Sciences, Qingdao Key Laboratory of Functional Membrane Material and Membrane Technology, No.189 Songling Road, 266101 Qingdao, China. [3]These authors contributed equally: Linfeng Li, Olena Vozniuk ✉ e-mail: olena.vozniuk@dupont.com; schueth@kofo.mpg.de

mild conditions, which normally require high temperatures and/or high pressures. Ball milling is the most frequently used implementation of mechanochemical reactions, and it has been used for materials synthesis[12–14] and organic synthesis[15–17]. Mechanochemical synthesis often avoids the use of solvents[18,19]; in addition, since the energy input stems from mechanical forces, external heating is often not required. Moreover, the collisions can improve mass transfer between reactants, which is highly favorable for reactions between solids[20]. Ball milling can also drive catalytic reactions. Even ammonia synthesis could be realized at room temperature and atmospheric pressure by ball milling, either in a discontinuous, two-step mechanism[21] or directly from the elements in continuous mode[22,23]. Charcoal gasification has recently been achieved by ball milling under much milder conditions than those of the traditional thermal process[24]. For CO oxidation[25] rates under milling increased by several orders of magnitude by in-situ ball milling compared to conventional plug-flow conditions, and in PROX, unusual selectivities were observed[26]. In a gas–solid reaction, methane chlorination by trichloroisocyanuric acid could also be achieved with a competitive reaction rate by ball milling under milder conditions and with higher selectivity than the thermal condition[27]. Depolymerization of polyethylene[28], Polyethylene terephthalate[29,30] and other plastics[31] can also be reached by ball milling.

Inspired by findings that ball milling can promote reactions between gases and solid materials, we explored ball milling for the hydrogenation of solid, carbon-containing substrates, most notably various polymers. Substrates (Supplementary Table S1) converted include different types of polymers, biomass, coal, activated carbon, and even diamond powder, which could be hydrogenated, partly at full conversion, to light hydrocarbons ($C_1$–$C_4$).

## Results

### Hydrogenation of different carbon substrates

At room temperature, ball-milling hydrogenation was carried out in a planetary ball mill where shear forces are the primary mode of energy transfer, supplemented by impact forces[32]. Ten 10 mm stainless steel balls were deployed in the milling jar (Supplementary Fig. S1). Three hundred milligrams γ-$Al_2O_3$ and 200 mg iron (Fe) powder as catalysts and 50 mg corresponding carbon substrates (polyethylene (PE), polyethylene terephthalate (PET), spruce wood, anthracite, hard coal, activated carbon (AC), diamond or $Fe_3C$) were added. It should be noted that although the molar amount of iron is higher than that of the carbon in the carbon substrate, iron is still referred to as catalyst, since it does not participate in the reaction as a reagent, but facilities the conversion. The milling jar was pressurized with gaseous hydrogen at 170 bar as the hydrogen source. After the reaction, the gas products were collected into a gas bag and then injected into a GC equipped with FID for quantitive analysis. The reaction mixtures left in the jar were collected and sent for element analysis to determine the carbon content. Both sets of results were combined to determine the full carbon balance. More details are given in the Methods section.

Surprisingly, all carbon substrates listed in Fig.1 (except PE) could be deeply hydrogenated at high conversion. The conversion of PE, however, remained at a relatively low level, even after extending the milling time from 7 to 21 h. In contrast, conversion of other carbon substrates increased from about 20–30% to about 90–99% upon extending milling time from 7 to 21 h (Fig. 1b). The low conversion of PE can be attributed to a combination of its physical resistance to abrasion and impacts, its relative softness and tendency for plastic deformation, dissipating mechanical energy more efficiently than the other substrates, and chemical inertness of the monotonous stable C–C bonds[33]. Methane is the main product in the hydrogenation of all substrates. $C_2$–$C_4$, only as alkane compounds, have also been detected. After 7 h of milling, the selectivity for methane always was above 80%. When extending the milling time to 21 h, the selectivity to methane was even higher at above 90%, and the selectivities for $C_2$–$C_4$ were reduced

or even fell below the detection limit (Fig. 1c, d and Supplementary Tables S2 and S3). This indicates that $C_2$–$C_4$ compounds generated at shorter milling times are hydrogenolytically cleaved with extended reaction time. In order to verify this hypothesis, butane ($C_4$) alone was used as the substrate. After milling for 7 h under the conditions of hydrogenation of solid substrates, about 10% butane conversion was observed, with methane as the main product, and also $C_2$ and $C_3$ could be detected, which proves the hydrogenolytic cleavage of butane. The carbon substrates used in this study contain different types of C–C bonds and also differ in heteroatom content and substitution. The hydrogenation of the different substrates, however, is undiscriminating. Even diamond powder and $Fe_3C$, in which carbon is bound to a metal atom, could be hydrogenated by this ball-milling hydrogenation method (Supplementary Fig. S5). The conversion of diamond reached 23% after 21 h milling. Diamond methanation via the thermal pathway requires at least a pressure of 0.5 GPa of hydrogen at 400 °C or 2 GPa at 300 °C[34]. Compared to the conditions of the ball-milling hydrogenation/depolymerization (no external heating and 170 bar (0.017 Gpa)), this again demonstrates that ball milling can drive reactions under much milder conditions.

### The effects of ball-milling parameters

During ball milling, kinetic energy from the milling balls can be inelastically transferred to the solids, leading to chemical and physical transformations. The initiation of a mechanochemically induced reaction and the extent is strongly dependent on milling parameters, such as rotation speed (alternatively, the milling frequency), filling ratio and ball-to-powder ratio, milling materials, and the size, number and combination of balls, among others[35]. Exploring in detail the combination of all these parameters would lead to a number of combinations, which is impossible to study. However, in order to obtain some insight into the crucial parameters, several of them were investigated in an exploratory fashion. The effect of the rotational speed was compared (Supplementary Fig. S3), and from the results it is evident that a minimum mechanical impact is needed to initiate the reaction; 300 rpm was the lowest rotational speed at which conversion could be detected in the setup used, albeit at a very low level. At 450 rpm and 600 rpm, activated carbon was mechanochemically converted to hydrocarbons to an appreciable extent: at 600 rpm, conversion levels were reached after 7 h, which required 21 h at 450 rpm. This clear positive correlation of rotational speed and conversion incidentally proves that the mechanical forces from the collisions of balls drive this hydrogenation reaction. When using bigger balls, i.e. three 10 mm balls, two 12 mm balls and one 15 mm ball, six in total (total mass is only 2% higher), the conversion dropped from 17% to 11% (Supplementary Fig. S4). In mechanochemistry, such changes are often difficult to interpret, but here a lower collision frequency caused by six instead of ten balls could be the reason. Ten smaller balls might also result in better mixing and a more homogenous distribution of impact energy than six bigger and differently sized balls.

### The radical mechanism

The fact that essentially any carbon-containing material seems to be hydrogenated to light hydrocarbons under ball milling is highly significant, both for its practical implications and for fundamental reasons, since such broad reactivity in hydrogenations at fairly mild conditions for solid substrates, compared to earlier attempts, which focus on one class of substrates, often under harsh conditions[36–39], had never been observed. It thus appeared interesting to explore the chemical mechanism behind this reaction. Based on scattered reports in the literature and the assumption that reactivity of such a broad range of substrates should not be caused by a highly selective reaction pathway, we decided to start the exploration with a radical mechanism as a working hypothesis. It had previously been observed that mechanical forces applied to both polymers or coal could generate

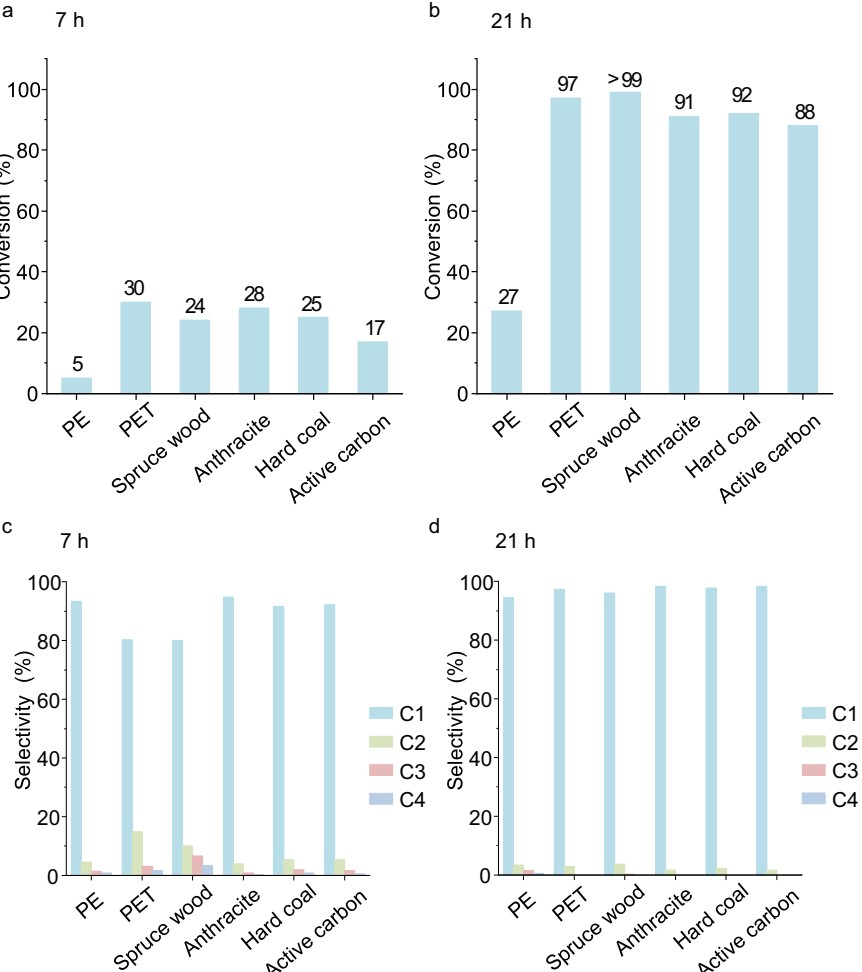

**Fig. 1 | Conversion and selectivity of ball-milling hydrogenation of different carbon substrates. a** Conversion of different carbon substrates after milling for 7 h. **b** Conversion of different carbon substrates after milling for 21 h. **c** Selectivity to $C_1$–$C_4$ after milling for 7 h. **d** Selectivity to $C_1$–$C_4$ after milling for 21 h. Reaction conditions: 450 rpm, 170 bar of $H_2$, 10 10 mm stainless steel balls, 50 mg of carbon substrates, 200 mg of Fe, 300 mg of $\gamma$-$Al_2O_3$. Conversion and selectivity data based on $C_1$–$C_4$ as described in the "Methods" section. PE is polyethylene. PET is polyethylene terephthalate. Source data are provided as a Source Data file.

free radicals[40,41]. The high reactivity of radicals could potentially account for the high efficiency of this mechanocatalytic system.

In order to obtain more insight into the role of possible radicals, we conducted hydrogenation experiments by adding the radical scavenger 2,2,6,6-tetramethylpiperidin-1-oxyl (TEMPO) to the normal reaction mixture/atmosphere. As shown in Fig. 2a, introducing TEMPO into the system decreased conversion of activated carbon from 17% to 8%. This would be in agreement with the function of a radical scavenger: it would capture radicals and thus prevent propagation of the radical reactions by eliminating the active radical species and forming stabilized radicals, which are intrinsically less reactive. The products left in the mill after the experiments were analyzed by High-Resolution Mass Spectrometry (HR-MS). TEMPO-trapping products, $C_1$-$C_5$ chains linked to TEMPO, were detected (Supplementary Table S4 Entry 1). A similar TEMPO radical trapping experiment was carried out during milling of polyethylene (PE). PE is a better defined substrate than the other solids studied, and thus conclusions can be drawn more straightforwardly. Also the hydrogenation reaction of PE was strongly suppressed by TEMPO (Fig. 2b), and $C_1$-$C_6$ chains linked to TEMPO were detected (Supplementary Table S4 Entry 2). This again supports the notion that the reaction proceeds via radicals. $CH_3$-TEMPO showed by far the highest intensity signal in HR-MS experiments (precise quantification is not possible in this system), which indicates that $CH_3\bullet$ is the most abundant radical species. This is in line with the high selectivity of $CH_4$ that forms through the binding of a $CH_3\bullet$ radical to a

hydrogen atom from a hydrogen molecule. Based on the TEMPO-trapping experiment, the mechanical force exerted upon carbon-containing substrates leads to active radicals, as had already been observed in the earlier studies described above. The final alkane products form through the combination of produced alkyl radicals and atomic hydrogens over metallic surfaces which usually accounts for $H_2$ dissociation in thermal catalytic hydrogenation reactions[42,43]. $H_2$ dissociation by mechanocatalysis should be feasible, on account of the reported work of ammonia synthesis[21–23] and hydrogenolysis of benzyl phenyl ether[44] under ambient hydrogen pressure, where $H_2$ dissociation is indispensable. Moreover, hydrogen dissociation also occurs on the metals used even without mechanical forces.

Besides radical generation by mechanical force, also high temperatures could lead to radical generation. When polymers are subjected to high-temperature processes, for instance pyrolysis and combustion[45–47], radicals can also be observed. The temperature of the exterior wall of the jar was monitored (Fig. 2d) by a wireless thermal couple (Supplementary Fig S1d, f) throughout the reaction. With the heating caused by continuous mechanical impact, the temperature increased to 49 °C after 8 h. This moderate increase in temperature is not expected to lead to substantially increased radical formation alone. However, hot spots at impact points between balls or between balls and walls have been discussed as one reason for altered reaction behavior in ball mills for decades, without being conclusively proven to be the origin[48]. In this study,

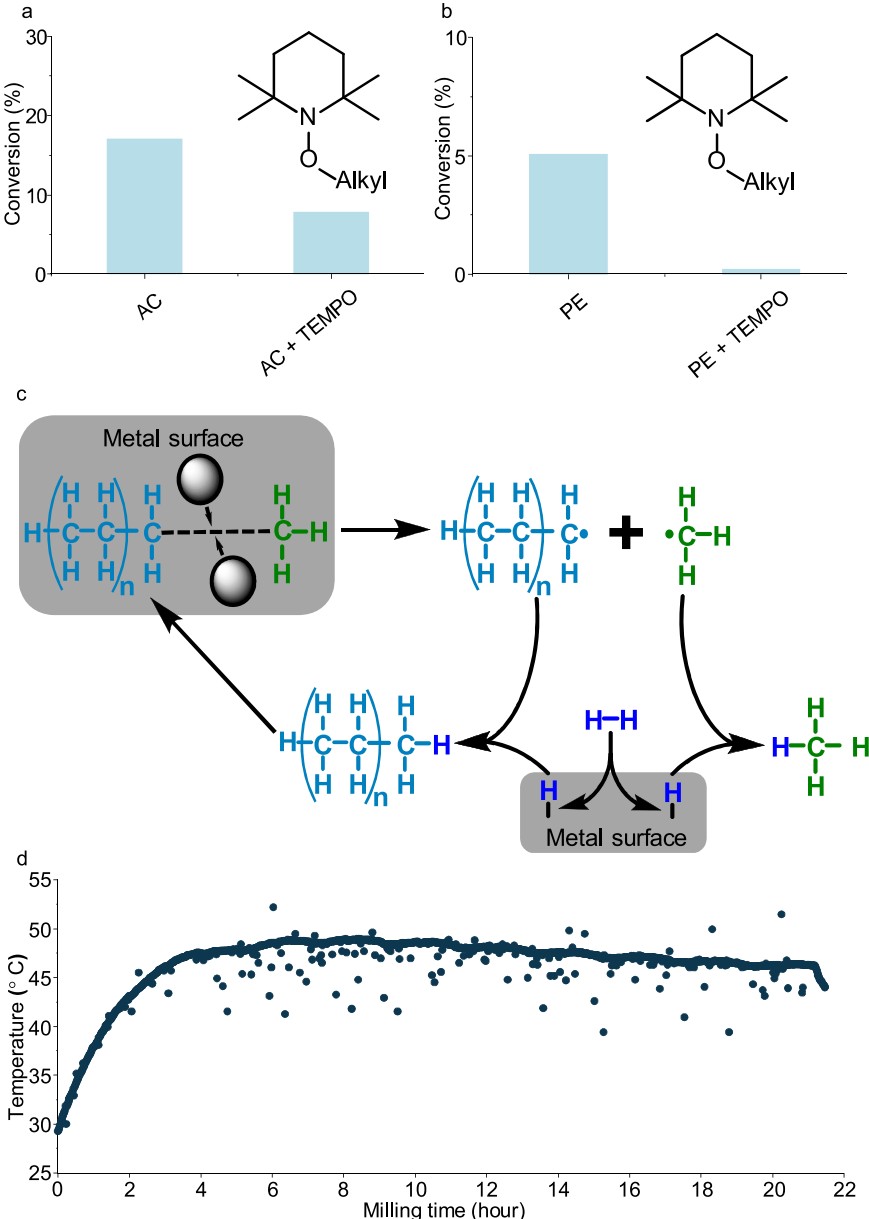

**Fig. 2 | TEMPO-trapping experiments and proposed mechanism. a, b** TEMPO-trapping experiments. Reaction conditions: 450 rpm, 7 h of milling time, 170 bar of $H_2$, 10 10 mm stainless steel balls, 50 mg of activated carbon (AC) or polyethylene (PE), 200 mg of Fe, 300 mg of γ-$Al_2O_3$, 25 mg of TEMPO. **c** Reaction mechanism. Alkyl radicals are first generated by mechanical forces. Then alkyl radicals react with surface atomic hydrogens which are from dissociative adsorption of $H_2$ over metal surface. **d** Temperature profile of exterior wall of the jar throughout the reaction. Reaction conditions: 450 rpm, 21 h of milling time, 170 bar of $H_2$, 10 10 mm stainless steel balls, 50 mg of activated carbon, 200 mg of Fe, 300 mg of γ-$Al_2O_3$. Source data are provided as a Source Data file.

the existence of hot spots, which could lead to radical formation, is a possibility, which cannot be excluded.

### The function of metal and support

We further compared the performance of different common hydrogenation catalysts, including ruthenium (Ru), cobalt (Co), nickel (Ni) and copper (Cu) (Fig. 3). Ru, Co, Fe, and Ni are quite similar in activity, with Co slightly better performing. However, Cu is not active under the conditions explored, the conversion is essentially zero. The energy barrier of homolytic $H_2$ dissociation over Cu is higher than those over other tested metals[43,49–51], but Cu with active sites for activation of $H_2$ is still employed in many hydrogenation reactions[52]. The total inactivity of Cu in this reaction leads to speculation that metal could take another crucial functional role, besides activation of $H_2$. Based on the mechanism

proposed, producing highly active alkyl radicals can be the potential step where metal plays a key role.

To explore the function of metal in the step of producing alkyl radicals, we conducted TEMPO-trapping experiments under argon atmosphere rather than hydrogen, so that alkyl radicals could be generated, but could not react further with hydrogen. PE was employed as the carbon substrate due to its well defined alkyl chain structure. Twenty five milligrams TEMPO was added to capture generated radicals. After milling for 7 h with Fe and γ-$Al_2O_3$, the remained products were analyzed by HR-MS, in which alkyl-linked TEMPO was detected (Fig. 4a table and Supplementary Table S4 Entry 3). The same TEMPO-trapping results were obtained when only Fe was added without γ-$Al_2O_3$ (Fig. 4a table and Supplementary Table S4 Entry 5). In contrast, when samples were milled with only γ-$Al_2O_3$ as additive in absence of Fe, no alkyl-linked TEMPO could be detected (Fig. 4a table

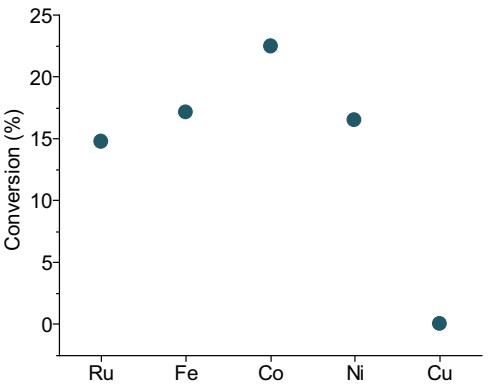

**Fig. 3 | Conversion of activated carbon by using Ru, Fe, Co, Ni and Cu.** Reaction conditions: 450 rpm, 7 h of milling time, 170 bar of H₂, 10 10 mm stainless steel balls, 50 mg of activated carbon, 200 mg of metal, 300 mg of γ-Al₂O₃. Source data are provided as a Source Data file.

and Supplementary Table S4 Entry 4). Thus, the metal seems to be essential for alkyl radical generation in this system.

The tentative understanding of how metal functions in the radical generation process could be enlightened by the performance of different metals (Fig. 3). Among metals tested, Ru, Co and Fe have good performance in Fischer-Tropsch reaction, where $C_{n+}$ species are stabilized and grow to longer chains. Ni is a bad catalyst for the Fischer-Tropsch reaction due to much stronger hydrogenation ability, leading mostly to $CH_4$. Cu is only employed for methanol synthesis without coupling function, unlike rhodium which also produces ethanol[53]. The adsorption energy of $C_2$ species over these five metals has been compared[54]. Over Cu, the adsorption energy is much less negative than those of Ru, Fe, Co and Ni, indicating Cu has the weakest ability to adsorb $C_2$ species. It should also be noted that the adsorption energies for Ru, Fe, Co and Ni are very similar. This is in line with the activity of this ball-milling hydrogenation, where Cu had no activity. Another TEMPO-trapping experiment was conducted by using Cu with γ-Al₂O₃ under argon atmosphere (Fig. 4a and Supplementary Table S4 Entry 6).

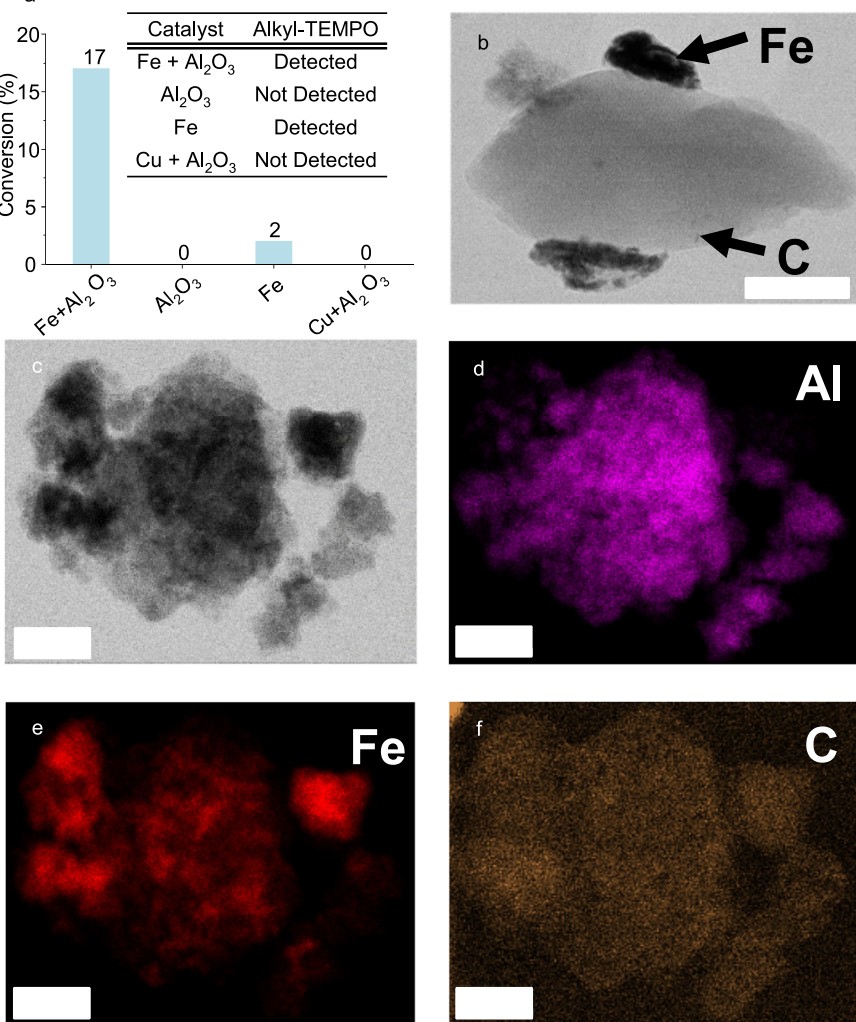

**Fig. 4 | The function of Fe and γ-Al₂O₃. a** Conversion of activated carbon by respectively adding 200 mg of Fe and 300 mg of γ-Al₂O₃, only 300 of mg γ-Al₂O₃, or only 200 mg of Fe, or 200 mg of Cu and 300 mg of γ-Al₂O₃. Reaction conditions: 450 rpm, 7 h of milling time, 170 bar of H₂, 10 10 mm stainless steel balls, 50 mg of activated carbon. Inset table: TEMPO-trapping experiments by respectively adding 200 mg of Fe and 300 mg of γ-Al₂O₃, only 300 of mg γ-Al₂O₃, or only 200 mg of Fe, or 200 mg of Cu and 300 mg of γ-Al₂O₃. Reaction conditions: 450 rpm, 7 h of milling time, 50 bar of Ar, 10 10 mm stainless steel balls, 50 mg of PE, 25 mg of TEMPO. **b** STEM image of the reaction mixture after hydrogenation of activated carbon by adding 200 mg of Fe, the scale bar represents 200 nm. **c** STEM images of the reaction mixture after hydrogenation of activated carbon by adding 200 mg of Fe and 300 mg of γ-Al₂O₃, the scale bar represents 100 nm. **d–f** EDX mapping image (the same frame as in **c**) of reaction mixture of hydrogenation of activated carbon by adding 200 mg of Fe and 300 mg of γ-Al₂O₃, the scale bar represents 100 nm. Source data are provided as a Source Data file.

No $C_2–C_6$ chains linked to TEMPO were detected and $CH_3$-TEMPO cannot be rigorously identified due to a very weak signal in HR-MS. Compared with other TEMPO-trapping experiments by using Fe (Fig. 4a and Supplementary Table S4 Entries 3 and 5), this could indicate the much poorer function of Cu in radical generation process. From these points one may hypothesize that strong adsorption of these carbon-based polymers over metals possibly is necessary for radical generation.

If the metal is an essential species, it would be expected that the hydrogenation would be improved by high dispersion of metal, and the γ-$Al_2O_3$ could serve as a dispersant for the metal, similar to previous observations on the synthesis of supported nanoparticles by ball milling[14,55]. The hydrogenation of activated carbon with Fe as additive was thus compared in presence and absence of γ-$Al_2O_3$ (Fig. 4a). As expected, the conversion dropped from 17% to 2%, when no γ-$Al_2O_3$ was added. Characterization with scanning transmission electron microscopy (STEM) (Fig. 4c) and energy-dispersive X-ray (EDX) (Fig. 4d–f) mapping of reaction mixtures after milling showed the well distributed carbon substrate and Fe in close association with the γ-$Al_2O_3$ phase, if both γ-$Al_2O_3$ and Fe were present during milling. In contrast, when γ-$Al_2O_3$ was absent, there is a clear separation of the Fe-catalyst and the carbon substrate, with rather big Fe particles (Fig. 4b). This suggests that the addition of γ-$Al_2O_3$ as milling additive aids in more homogeneously distributing the carbon substrate and Fe by forming a solid matrix. γ-$Al_2O_3$ alone is inactive for this hydrogenation, since with only γ-$Al_2O_3$ there was no detectable conversion.

Noticeably, if the milling results and those of the TEMPO-trapping experiments are compared (Fig. 4a), it is obvious that conversion is only achieved when alkyl-TEMPO is detected. This is in line with the radical mechanism. Likewise (Fig. 4a), any substrate conversion and also alkyl-TEMPO were only observed in the presence of Fe, emphasizing its functional role in the step of radical generation in this hydrogenation reaction.

### The performance of MgO as support
In order to further explore the potential of this reaction, we tested different supports and hydrogenation/depolymerization under lower pressure. MgO was found to be a highly suitable support among the supports screened (Fig. 5a). After 7 h of milling, 83% conversion of AC can be reached over MgO. In contrast, the conversions over the other supports were all below 20%. Support effects are diverse in heterogeneous catalysis[56]. To fully understand the support effects in this reaction, much more extensive studies are required which is beyond the scope of this work. However, these initial data suggest that there is substantial upside potential for these systems.

Different hydrogen pressures were compared by using MgO as support (Fig. 5b). The conversion did not change significantly when the pressure was reduced from 170 bar to 20 bar. Upon further decreasing the hydrogen pressure to 5 bar, the conversion dropped to 59%(under these conditions, 5 bar hydrogen pressure would be sufficient to just convert the full amount of carbon present).

Under a hydrogen pressure of 20 bar, the stability of the Fe-MgO catalyst was tested (Fig. 5c). In the first round 50 mg of activated carbon was added. In each subsequent round, an equal amount of activated carbon was newly added to replenish the converted activated carbon from the previous round, ensuring a consistent initial carbon amount throughout each round. The conversion dropped to 44% in the fifth round. This could be due to different factors: the surface of the catalyst may be poisoned by impurities or side products of the reactions, alternatively, the easy-to-depolymerize carbon may react first, so that after each repetition upon replenishing only the converted carbon, the difficult-to-polymerize carbon remains in the system. A detailed study of the deactivation is subject of current investigations.

## Discussion
In summary, we have discovered that a broad variety of solid carbon-containing substrates can be fully hydrogenated at room temperature to small hydrocarbon molecules in a mechanocatalytic reaction with late first row transition metals as suitable catalysts. The system shows a consistently good hydrogenation efficiency for common carbon substrates, such as different polymers, biomass, and different coal grades. To the best of our knowledge, there is no other catalytic method for hydrogenation of such difficult-to-activate solid carbon substrates at such a mild temperature with high efficiency. The good hydrogenation performance of this system is crucially dependent on both physical and chemical factors. The physical origin of activity in our ball-milling system is the transfer of mechanical energy, with a clear correlation between intensity of energy input and hydrogenation effectiveness. With further exploration of such systems, most probably better ball-milling conditions than those employed here can be identified, possibly also with altered product distribution. Optimization of conditions could possibly be supported by simulation[57,58]. Chemically, all

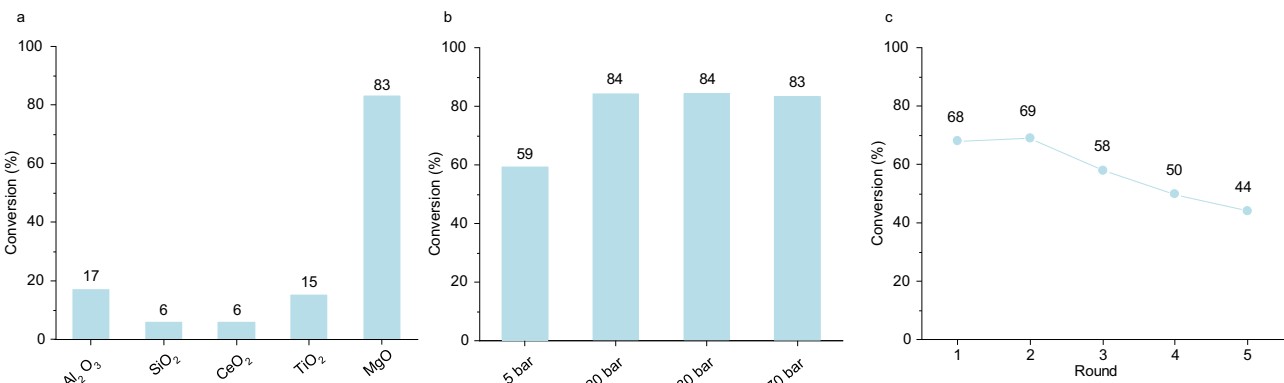

**Fig. 5 | The performance of MgO as support. a** Conversion of activated carbon by different supports. Reaction conditions: 450 rpm, 7 hours of milling time, 170 bar of $H_2$, 10 10 mm stainless steel balls, 50 mg of activated carbon, 200 mg of Fe, 300 mg of γ-$Al_2O_3$, $SiO_2$, $CeO_2$, $TiO_2$ or MgO. **b**, Conversion of activated carbon with different initial hydrogen pressure. Reaction conditions: 450 rpm, 7 h of milling time, 5, 20, 80 or 170 bar of $H_2$, 10 10 mm stainless steel balls, 50 mg of activated carbon, 200 mg of Fe, 300 mg of MgO. (For easier gas products collection, in the reaction under 5 bar of $H_2$, in addition to 5 bar of $H_2$, 25 bar of Ar was charged. The total pressure is 30 bar.). **c** Stability of Fe–MgO catalysts. Reaction conditions: 450 rpm, 6 h of milling time, 20 bar of $H_2$, 10 10 mm stainless steel balls, 200 mg of Fe, 300 mg of MgO. In the first round, 50 mg activated carbon was added. In each subsequent round, an equal amount of activated carbon was newly added to replenish the converted activated carbon from the previous round, ensuring a consistent initial carbon amount throughout each round. Source data are provided as a Source Data file.

experimental evidence points to an essentially radical-driven mechanism, where the metal catalysts play a crucial role. However, the exact reaction pathways still need to be elucidated, for which operando measurements for the detection of the radicals would certainly be very helpful. However, in spite of first successful attempts[59,60], studying in-situ or operando mechanochemical processes is very difficult, due to the moving system and the robust walls of at least a few millimeters in thickness. This leaves a great challenge for further study, in addition to the exploration of the usefulness of such approaches under practically relevant conditions on interesting substrates, such as polymers or different coal grades.

## Methods

### Catalyst, carbon substrates and chemicals

Iron (Fe) powder from Sigma-Aldrich, powder (fine), ≥99% purity; cobalt (Co) powder from Sigma-Aldrich, powder, <150 µm, ≥99.9% trace metals basis; nickel (Ni) powder from Goodfellow, powder, <45 µm, >99.5% purity; copper (Cu) powder from Sigma-Aldrich, powder (dendritic), <45 µm, 99.7% trace metals basis; ruthenium (Ru) powder from Sigma-Aldrich,−200 mesh, 99.9% trace metals basis; aluminum oxide (γ-Al$_2$O$_3$) from Alfa Aesar, 2.5 micron powder, S.A. 100–150 m$^2$/g, 99.997% (metal basis); magnesium oxide (MgO) from Sigma-Aldrich, −325 mesh, ≥99%, trace metals basis, calcined at 750 °C (10 °C min$^{-1}$, 24 h) prior to use; silica (SiO$_2$) from Sigma-Aldrich, high-purity grade (Davisil Grade 62), pore size 150 Å, 60–200 mesh; titanium(IV) oxide (TiO$_2$) from Sigma-Aldrich, nanopowder, <25 nm particle size, 99.7% trace metals basis; cerium(IV) oxide (CeO$_2$) from Sigma-Aldrich, powder, 99.995% trace metals basis; polyethylene (PE) from Sigma-Aldrich, powder, spectrophotometric grade; polyethylenterephthalat (PET) from Goodfellow, copolymer, semi crystal-line, <300 µm; spruce wood chips from J. Rettenmaier & Söhne;. anthracite from coal mine Ibbenbüren; hard coal from coal mine Osterfeld; activated carbon powder from NORIT; (2, 2, 6, 6-Tetramethylpiperidin-1-yl)oxyl (TEMPO) from Sigma-Aldrich, 98% purity;

### Carbon substrates preparation

Spruce wood chips, anthracite and hard coal were ground into fine powders using a Retsch CryoMill. (Supplementary Fig. S6). The milling process was divided into two steps: pre-cooling and freeze milling. In both steps, the milling vessel was continuously cooled with liquid N$_2$ from the integrated cooling system. Before freeze milling started, the milling vessel had already been cooled down by liquid N$_2$. The milling parameters used: t (pre-cooling) = 10 min, oscillation frequency (pre-cooling) = 5 Hz, t (freeze milling) = 30 min, oscillation frequency (freeze milling) = 25 Hz, one 15 mm stainless steel ball, 2 g of carbon substrate.

### Determination of carbon content in carbon substrates and reaction mixtures

Elemental analysis was performed by Mikroanalytisches Laboratorium Kolbe (c/o Fraunhofer Institut UMSICHT, Osterfelderstraße 3, D46047 Oberhausen). The analysis was performed with the help of a CHNOS analyzer of the brand ELEMENTAR model Vario Micro Cube.

### General procedure for ball-milling hydrogenation

All the experiments were carried out in Fritsch Pulverisette 6 planetary ball mill. The high-pressure steel (type 1.4571) milling jar (Supplementary Fig. S1) was manufactured in our workshop. The milling jar was equipped with a Teflon-lid to prevent powder escaping. As a standard procedure, powders of a catalyst and a C-source were added into the milling jar where the balls had been placed. The milling chamber was then covered by the Teflon-lid, followed by closing the milling jar by its lid, which was tightened by a torque wrench at 20 Nm. Before pressurizing with pure hydrogen (P$_{H2}$ = 170 bar), the milling vial was evacuated three times with the following sequence of steps: Ar

loading/vacuum applied. Pressure for calculations was taken after the reaction by connecting the valve to a pressure sensor connected to a pressure reading system. The obtained residuals of PET and spruce wood (which had conversion close to 100%) after reaction were measured by elemental analysis to determine the carbon content, and the carbon balance was calculated as follows:

$$\text{Carbon balance} = \frac{\text{moles of carbon in C}_1 - \text{C}_4 + \text{moles of carbon in residuals}}{\text{moles of carbon in starting material}} \times 100\%$$

(1)

The obtained carbon balance for PET and spruce wood are 104% and 107%, respectively.

### Characterization of gas products

**GC-FID.** Gas phase hydrocarbons (C$_1$–C$_4$) were analyzed by GC-FID (Agilent 6850) equipped with an Agilent column (DB-624, 30 m × 0.320 mm × 1.80 µm). As a standard procedure, gaseous products were collected from a pressurized milling vial into a gas bag (purchased from Sigma-Aldrich, type: Supel-Inert Multi-Layer Foil, maximum volume 0.6 L, Screw Cap Valve (SCV)); manual injection was done with a gas tight syringe (Hamilton GASTIGHT 1750, max volume of 500 µL). Volume of the calibration gas as well as the volume of the analyzed products was fixed to 400 µL. Three injections were repeated for each analysis and an average was taken. The error among the three injections is less than 5%.

**Calculations.** Quantification of C$_1$–C$_4$ hydrocarbons was done based on the moles of obtained products from GC-FID:

$$\text{Conversion} = \frac{\text{moles of Carbon in C}_1 - \text{C}_4}{\text{moles of carbon in starting material}} \times 100\%$$

(2)

$$\text{Selectivity} = \frac{\text{moles of C}_n}{\text{moles of carbon in C}_1 - \text{C}_4} \times 100\%$$

(3)

**HR-MS.** Thermo Scientific Q Exactive plus.

**STEM.** Hitachi HD-2700 CS-corrected dedicated STEM 200 kV, Cold FEG.

**EDX.** EDAX Octane T Ultra W 200 mm2 SDD TEAM-Software.

## Data availability

The data generated in this study are all available within the Source Data file, as well as the Supplementary Information file. Source data are provided with this paper.

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

## Acknowledgements

We gratefully thank the mass spectrometry laboratory at the Max-Planck-Institut für Kohlenforschung led by PD Dr. Wolfgang Schrader for HR-MS measurement. We gratefully thank the finemechanics department at the Max-Planck-Institut für Kohlenforschung led by Wolfgang Kersten for building the milling equipment. We greatly acknowledge the High-pressure lab at the Max-Planck-Institut für Kohlenforschung led by Lars Winkel. We thank department of electron microscopy at the Max-Planck-Institut für Kohlenforschung led by Prof. Dr. Christian W. Lehmann for STEM and EDX measurement. We thank Mikrolab Kolbe for the elemental analysis. We thank Kai Jeske, Zihang Qiu, Özgül Agbaba, Jiyao Yan and Phil Hesse who is from the department of chromatography at the Max-Planck-Institut für Kohlenforschung led by Dr. Philipp Schulze for the help for GC measurement. We thank Jan Ternieden from department of powder diffraction and surface spectroscopy at the Max-Planck-Institut für Kohlenforschung led by Prof. Dr. Claudia Weidenthaler for XRD measurement.We thank JiKai Ye, Jacopo de Bellis, Steffen Reichle, Frederik Winkelmann and Bolun Wang for the help with ball-milling equipment and process. We thank Matthias Haenel for helpful discussion.

## Author contributions

Conceptualization, F.S., O.V. and L.F.L.; Methodology, L.F.L., O.V., P.L., Z.W.C., M.F. and F.S.; Investigation, L.F.L., O.V., P.L. Z.W.C.; Writing – original draft, L.F.L., Z.W.C, O.V., and F.S.; all authors worked on the final version of the manuscript; Supervision, F.S.

## Funding

## Competing interests

The authors declare no competing interests.

## Additional information

**Publisher's note** Springer Nature remains with regard to jurisdictional claims in published maps and institutional affiliations.

