## [Peer Review File · Nature Communications]

Hydrogenation of different carbon substrates into light hydrocarbons by ball millingREVIEWER COMMENTS

Reviewer #1 (Remarks to the Author):

The important result of this article is the demonstration that many sources of carbon, such as anthracite, carbon, spruce and PET, can be converted into mainly methane in high yields, without heating, thanks for mechanochemistry. This reaction proceeds at high pressure and relies on Fe with alumina support as reagent. This work is innovative and brings a lot of new ideas to both the question on waste recovery and carbon cycling, as well as fundamental principles in mechanochemistry.

Based on the importance of the points above, This article of suitable for publication in Nature Communication. That being said, I recommend major revisions to address two main questions. A. The authors should provide more information on the reaction conditions and gain more insight into how materials evolve during the reaction. B. Also, they should improve the manuscript to bring more clarity over the exact nature of the reaction itself. For people unfamiliar with mechanochemistry, and even for those you know it better, it is very hard to have an understanding of what researchers did upon a single read because of the lack of information in the text.

A1. Authors provide an excellent background in the introduction on the thermal version of the studies reaction. As the reaction is milled in a planetary mill with fair stiff material and for hours, the reaction is likely to go up in temperature. There are two dimensions to this question. One is that the macroscopic temperature must be measured and reported. This is easy to do and it would be important for reproductibility.

A2. The second aspect is hot spots. Because of the nature of this reaction, it is possible that hot spots can be involved. This is obviously much harder to measure, but it could be discussed in the context of other articles in the field, for instance: <https://doi.org/10.1016/j.cej.2019.122954>

A3. The TEM study of the way particles of Fe evolve during the reaction is very limited, while authors point themselves at the centrality of this point. They demonstrated that alumina was needed and that it served as a support. But I think further studies would be essential to understand how particles evolve from the beginning to the end of the reaction. Average particle sizes should be provided. It should not require a lot of work and authors may already have enough images to do that. It would really help the conversation.

A4. Based on the idea that particles need to undergo comminution with the alumina, is the reaction faster if researchers pre mill alumina and Fe? or reuse a catalyst for a second round?

A5. Fig2, mechanism. I understand and am convinced by the evidence radicals are present, but I am less convinced that the first step is correct. It suggest mechanochemistry alone is responsible for the radical formation, when Alumina alone reaction disproves it. I think the role of Fe is more important and should not be understated in the mechanism.

A6. l. 144 145: Please discuss more H₂ dissociation. The pressure is very high and will certainly help. References were are only in a thermal context. Can mechanochemistry help?

A7. PET and wood degradation will provide water as by product. did you detect it? can you comment on that?

A8. PET chemical depolymerisation was reported and should be cited, even if, of course, the mechanism

here is very different. <https://doi.org/10.1002/cssc.202002124> & <https://doi.org/10.1021/acssuschemeng.2c03376>

B1. While these details on reaction conditions are provided in captions (in part) and experimental section (although the reagents masses are not provided there but in tables later), they are not well explained in the text. It is essential to have the reaction explained at the beginning of the discussion (l. 67). This includes giving the number, size and materials of the balls, the H₂ pressure, and the respective mass of the reagents.

B2. The full list of carbons tested must be provided as well in the text (l. 68)

B3. While diamond powder is evoked as a possible C source, it's not shown in results, for instance Fig. 1. Please add it and discuss it.

B4. l. 64. Fe is added at 200mg, alumina 300 mg and the carbon to convert 50 mg. By definition Fe is not added in catalytic quantity. I understand why authors refer to it as a catalyst, but this denomination is misleading if not put in context. Please clarify this point.

B5. l.96 How is conversion measured. What did authors do to control for mass balance? How did they recover the solid part of left over C to measure it (distinguish from alumina/Fe). These are important points to understand the reaction, they should be added to the main text succinctly.

B6. It's not clear if H₂ is present during TEMPO test.

B7. l. 125, please replace substantially by actual values

Reviewer #2 (Remarks to the Author):

The hydrogenation of carbon-based solids into hydrocarbons is widely used in industrial applications, but it is a slow reaction since the strong carbon-carbon bonds in the carbon frameworks. This paper demonstrates this reaction can be proceeded efficiently at room temperature by mechanochemistry. The authors systematically investigated the broad range of different carbon substrates and the influence of different metal catalysts. Importantly, they took a deeper insight into the mechanism of mechanochemical hydrogenation. Using the radical scavenger, TEMPO, the proposed radical mechanism of mechanochemical hydrogenation is reliable and impressive, it could guide more exploration to catalytical reaction by novel approach. Overall, this manuscript will attract broad interest, and thus it is worthy to consider in Nat. Commun. after addressing following issues.

1. In the Extended Data Tables 2 and 3, please specify the C1-C4 products to clarify whether olefin or alkyne exists in the C2-C4 products.
2. For the products, the authors do not mention whether H₂ completely reacts. If there is unconsumed H₂ in the container after reaction, it should be considered in calculating the selectivity of products.
3. The supporting material plays an important role in the catalytic reaction, and therefore it is necessary to add contrast experiments based on different supporting materials such as CeO₂, MgO and SiO₂.
4. The authors display the high efficiency of the first round of mechanochemical hydrogenation, please add the catalytic stability tests of Fe/ γ -Al₂O₃ catalyst.

5. The reviewer notices the pressure of H₂ is 170 bar, this is harsh condition and one thinks this experimental condition could not be considered as a mild condition which mentioned in the manuscript. Also, the small amounts of carbon-based solids and extremely high pressure may not be substantial technical leap for scale-up mechanochemical hydrogenation of carbon-based solids. Thus, the reviewer suggests checking whether this reaction can be proceeded at a lower pressure of H₂ as well as bigger scale of reaction to still obtain high efficiency by mechanochemistry.

Reviewer #3 (Remarks to the Author):

The manuscript describes a mechanocatalytic process for converting different carbon-based solids methane and small amounts of higher hydrocarbons. The manuscript is quite interesting and should be published once the following issues are addressed:

1. It would be nice if the authors could discuss why PE is significantly less reactive than any of the other materials studied.
2. If butane conversion after 7 h of milling reaching only 10% the high methane yield from the solid feedstocks is not entirely explained by formation of intermediates and their subsequent methanation.
3. In lines 110 to 114, the authors refer to earlier attempts to produce light hydrocarbons from solid feedstocks in vague terms. They should provide specific examples and references.
4. Ruthenium has shown considerable activation for hydrothermal methanation of biomass. Thus, it would be interesting to include it in the comparison of metal catalysts.

Minor issues:

Line 81: methane should not be capitalized

Line 194: Typo in carbon-based polymers

We are very grateful to all reviewers' supportive comments and advice for this work. According to your advice, we have improved our work, as listed below. The answer to each point is colored by red. Those highlighted by yellow are the discussions added in the manuscript.

REVIEWER COMMENTS

Reviewer #1 (Remarks to the Author):

The important result of this article is the demonstration that many sources of carbon, such as anthracite, carbon, spruce and PET, can be converted into mainly methane in high yields, without heating, thanks for mechanochemistry. This reaction proceeds at high pressure and relies on Fe with alumina support as reagent. This work is innovative and brings a lot of new ideas to both the question on waste recovery and carbon cycling, as well as fundamental principles in mechanochemistry.

Based on the importance of the points above, This article of suitable for publication in Nature Communication. That being said, I recommend major revisions to address two main questions. A. The authors should provide more information on the reaction conditions and gain more insight into how materials evolve during the reaction. B. Also, they should improve the manuscript to bring more clarity over the exact nature of the reaction itself. For people unfamiliar with mechanochemistry, and even for those you know it better, it is very hard to have an understanding of what researchers did upon a single read because of the lack of information in the text.

A1. Authors provide an excellent background in the introduction on the thermal version of the studies reaction. As the reaction is milled in a planetary mill with fair stiff material and for hours, the reaction is likely to go up in temperature. There are two dimensions to this question. One is that the macroscopic temperature must be measured and reported. This is easy to do and it would be important for reproductibility.

- We modified the foundation for the milling jar to have a wireless thermal couple inserted on the internal wall (see the Figure below and Extended Data Fig 1 in the manuscript). After embedding the milling jar into the foundation. The thermal couple is in close contact to the outside wall of the milling jar. Through this, the temperature of the outside wall of the milling jar was measured throughout the reaction (see the Figure below). The discussion was combined with the answer to comment A2 (see below). The temperature profile measured with this thermocouple was inserted into Fig.2 as Fig. 2d. The temperature increase measured corresponds well to many studie¹⁻⁴ in which the global temperature in the milling vessel was determined by various means, i.e. increase in temperature of a few ten

Kelvin at most.

A2. The second aspect is hot spots. Because of the nature of this reaction, it is possible that hot spots can be involved. This is obviously much harder to measure, but it could be discussed in the context of other articles in the field, for instance: <https://doi.org/10.1016/j.cej.2019.122954>

- Hot spots are discussed as a possible contribution in many mechanochemical reactions, but the issue has not been clarified over more than 60 years. It is subject of an extensive research proposal of F. Schüth. We have added a corresponding discussion combining A1 and A2 in the text. Below is the discussion. (Line 170–181 in the manuscript)

Except radical generation by mechanical force, also high temperatures could lead to radical generation. When polymers are subjected to high temperature processes, for instance pyrolysis and combustion⁵⁻⁷, radicals can also be observed. The temperature of the exterior wall of the jar was monitored (Fig. 2d) by a wireless thermal couple (Extended Data Fig 1. d and f) throughout the reaction. With

the heating caused by continuous mechanical impact, the temperature increased to 49 °C after 8 hours. This moderate increase in temperature is not expected to lead to substantially increased radical formation alone. However, hot spots at impact points between balls or between balls and walls have been discussed as one reason for altered reaction behavior in ball mills for decades, without being conclusively proven to be the origin⁸. In this study, the existence of hot spots, which could lead to radical formation, is a possibility, which cannot be excluded.

A3. The TEM study of the way particles of Fe evolve during the reaction is very limited, while authors point themselves at the centrality of this point. They demonstrated that alumina was needed and that it served as a support. But I think further studies would be essential to understand how particles evolve from the beginning to the end of the reaction. Average particle sizes should be provided. It should not require a lot of work and authors may already have enough images to do that. It would really help the conversation.

- We agree that a detailed analysis of the development of particle sizes and distributions as well as the different phases would help to deeper understand the process, since particle size effects are common in many thermal/electrical/photo catalytic reactions, and different phases could have different reactivities. However, the presence of ferromagnetic iron particles in the catalyst powder hinders detailed analysis, since the iron particles are attracted by the magnetic field of the lenses in the TEM, which limits possibilities of analysis. These two TEM images below, with the size bar of 100 nm, are from mixtures after milling 200 mg Fe, 300 mg

Al₂O₃ and 50 mg active carbon under 170 bar hydrogen for 7(left) and 21 (right)hours, with little obvious difference.

7 hours

21 hours

- Moreover, we include above the XRD patterns of the solid after 7h and 21 h. As one can see, after 7 h there is a mixture of phases, which is difficult to fully assign quantitatively - the low signal-to-noise ratio also suggests the presence of substantial amounts of amorphous material. After 21 h, there is essentially iron, corundum, and iron aluminum oxide. A full analysis of the transformation of the solid in the jar and its correlation with reactivity at any given time is in progress, but far beyond the scope of this paper, since also the influence of the different

substrates would need to be analyzed.

A4. Based on the idea that particles need to undergo comminution with the alumina, is the reaction faster if researchers pre mill alumina and Fe? or reuse a catalyst for a second round?

- We first milled 200 mg Fe and 300 mg Al_2O_3 under Ar for 1 hour. 50 mg active carbon was then added into the jar. After 7 hours of milling under hydrogen, the conversion was 51%, compared to 17% without pre milling catalysts. Thus, the reaction rate is higher after the iron is dispersed over Al_2O_3 by pre milling, compared to starting with separate Fe and Al_2O_3 powders.

A5. Fig2, mechanism. I understand and am convinced by the evidence radicals are present, but I am less convinced that the first step is correct. It suggest mechanochemistry alone is responsible for the radical formation, when Alumina alone reaction disproves it. I think the role of Fe is more important and should not be understated in the mechanism.

- Yes, we agree. Metals play a key role in the first step to producing radicals. Below we summarize how we explored the metal's functional role in the reaction. For more details please see also line 183–224 in the manuscript.

We compared the performance of different metals: Ru, Fe, Co and Ni exhibited similar conversions; however, Cu, which is also active in other hydrogenation reactions, was inactive under the conditions used here. This suggests that some metals are participating in generating alkyl radicals, with copper being unable to facilitate this step

The metal's function in producing alkyl radicals was explored also by TEMPO trapping experiments. The TEMPO-trapping experiments were conducted under Ar so that the next step, hydrogenation of alkyl radicals, does not take place. As a result (Fig 4a table), only when

iron is present, the alkyl-TEMPO could be detected. Consequently, also only when iron is present does the hydrogenation proceed (Fig 4a histogram). Thus metals are essential for alkyl radical generation.

There is also literature support in the function of the metals in radical generation. The ability of the tested metals to adsorb carbon species is in line with the activity of metals in the reaction studied here. Ru, Fe, Co and Ni have similar ability to adsorb carbon species⁹. On the other hand, copper, which is inactive in this reaction, has the weakest ability to adsorb carbon species. Another TEMPO trapping experiment was conducted by using Cu with γ -Al₂O₃ under Ar and no alkyl-TEMPO signal could be detected (Fig .4a). This indicates the much poorer function of Cu in the radical generation.

While these experiments do not clarify exactly, how the metals, together with the mechanical forces, lead to formation of radicals, one can clearly see that the metal is necessary for radical generation, possibly through the adsorption of the substrates on the metal surface.

A6. 1. 144 145: Please discuss more H₂ dissociation. The pressure is very high and will certainly help. References were are only in a thermal context. Can mechanochemistry help?

- Ammonia synthesis as already mentioned in the context can be reached at ambient pressure by ball milling⁵⁻⁷. In a recent study, hydrogenolysis of benzyl phenyl ether (a model lignin molecule) also proceeds at ambient pressure through mechanochemistry⁸. H₂ dissociation at mild conditions can thus be realized by mechanochemistry - and the metals are also known to dissociate hydrogen also without using mechanical forces. However, deeper understanding, especially in the context of the substrates converted here, would require in depth studies, which are beyond the scope of this work. This point is discussed now in the manuscript as given below (Line 166-169 in the manuscript)

H₂ dissociation by mechanocatalysis should be feasible, on account of the reported work of ammonia synthesis¹⁰⁻¹² and hydrogenolysis of benzyl phenyl ether¹³ under ambient hydrogen pressure, where H₂ dissociation is indispensable. Moreover, hydrogen dissociation also occurs on the metals used even without mechanical forces.

A7. PET and wood degradation will provide water as by product. did you detect

it? can you comment on that?

- In addition to monitoring the carbon balance, we made substantial efforts to quantify the amount of water generated. This was done to better control the oxygen balance, gain insight into the role of oxygen in the reaction, and ensure its proper management.

The oxygen produced from the degradation of PET can react with hydrogen to form water or react with metals to form oxides, in addition, water may be formed directly in the depolymerization. In our case, it is more likely that water is formed. To confirm this, we attempted to identify and quantify the water content using GC-TCD or Karl-Fischer methods. However, the measured amount of water after reaction was only about half of the expected amount (10 mg as compared to 19 mg) There are two possible reasons for this:

i) The generated water may have been adsorbed onto the catalyst, formed hydrates, or react mechanochemically with the carbon surface to result in different types of surface oxygenates, making it difficult to recover/detect it completely.

ii) Due to the limited amount of water and the complex nature of water exchange between the water in the jar and the atmosphere (whether in a glove box or ambient atmosphere) after opening, we do not have an effective method to fully recover the water and provide an exact measurement of its amount.

However, the carbon balance does indicate that oxygen was removed in the final product, most likely in the form of water.

A8. PET chemical depolymerisation was reported and should be cited, even if, of course, the mechanism here is very different.

<https://doi.org/10.1002/cssc.202002124>

&

<https://doi.org/10.1021/acssuschemeng.2c03376>

- References has been added (Line 55-56 in the manuscript)

Depolymerization of polyethylene¹⁴, PET^{15,16} and other plastics¹⁷

can also be reached by ball milling.

B1. While these details on reaction conditions are provided in captions (in part) and experimental section (although the reagents masses are not provided there but in tables later), they are not well explained in the text. It is essential to have the reaction explained at the beginning of the discussion (l. 67). This includes giving the number, size and materials of the balls, the H₂ pressure, and the respective mass of the reagents.

- The details have been added, making the manuscript hopefully more readable. (Line 65 -72 in the manuscript)

Ten 10 mm stainless-steel balls were deployed in the milling jar

(Extended Data Fig. 1). 300 mg γ -Al₂O₃ and 200 mg iron (Fe) powder as catalysts and 50 mg corresponding carbon substrates (polyethylene (PE), polyethylene terephthalate (PET), spruce wood, anthracite, hard coal, active carbon (AC), diamond or Fe₃C) were added. It should be noted that although the molar amount of iron is higher than that of the carbon in the carbon substrate, iron is still referred to as catalyst, since it does not participate in the reaction as a reagent, but facilitates the conversion. The milling jar was pressurized with gaseous hydrogen at 170 bar as the hydrogen source.

B2. The full list of carbons tested must be provided as well in the text (l. 68)

- The full list has been provided. (Line 67–68 in the manuscript)

50 mg corresponding carbon substrates (polyethylene (PE), polyethylene terephthalate (PET), spruce wood, anthracite, hard coal, active carbon (AC), diamond or Fe₃C) were added.

B3. While diamond powder is evoked as a possible C source, it's not shown in results, for instance Fig. 1. Please add it and discuss it.

- We added the result and discussion in the main text as below. (Line 100–105 in the manuscript)

The conversion of diamond reached 23% after 21 hours milling. Diamond methanation via the thermal pathway requires at least a pressure of 0.5 GPa of hydrogen at 400 °C or 2 GPa at 300 °C.¹⁸ Compared to the conditions of the ball milling hydrogenation/depolymerization (no external heating and 170 bar (0.017 GPa)), this again demonstrates that ball milling can drive reactions under much milder conditions.

B4. 1. 64. Fe is added at 200mg, alumina 300 mg and the carbon to convert 50 mg. By definition Fe is not added in catalytic quantity. I understand why authors refer to it as a catalyst, but this denomination is misleading if not put in context. Please clarify this point.

- This point was clarified in the main text as below. (Line 69-71 in the manuscript)

It should be noted that although the molar amount of iron is higher than that of the carbon in the carbon substrate, iron is still referred to as catalyst, since it does not participate in the reaction as a reagent, but facilitates the conversion

B5. 1.96 How is conversion measured. What did authors do to control for mass balance? How did they recover the solid part of left over C to measure it (distinguish from alumina/Fe). These are important points to understand the reaction, they should be added to the main text succinctly.

- These have been added into the main text. (Line 72-76 in the manuscript)
We did not separate C from alumina/Fe, the mixtures were directly sent to Mikroanalytisches Laboratorium Kolbe, an independent company for element analysis, to determine the carbon content.

After the reaction, the gas products were collected to a gas bag and then injected into a GC equipped with FID for quantitative analysis. The reaction mixtures left in the jar were collected and sent for element analysis to determine the carbon content. Both sets of results were combined to determine the full carbon balance. More details are given in the Methods section.

B6. It's not clear if H₂ is present during TEMPO test.

- We modified the wording to clarify the conditions during the experiments with TEMPO. (Line 142-144 and 194-196 in the manuscript)

we conducted hydrogenation experiments by adding the radical scavenger 2,2,6,6-tetramethylpiperidin-1-oxyl (TEMPO) to the

normal reaction mixture/atmosphere.

we conducted TEMPO-trapping experiments under argon atmosphere rather than hydrogen, so that alkyl radicals could be generated, but could not react further with hydrogen.

Some more explanations only given here:

In TEMPO tests shown in Figure.2 a&b, H₂ was present. These experiment served to show, whether TEMPO could trap alkyl radicals and thus suppress the hydrogenation reaction.

In the TEMPO tests shown in Figure. 4 a/table, H₂ was absent. These experiments served to investigate the metal' s function in the first step, i.e. of radical generation. To avoid the further hydrogenation step, we conducted these TEMPO tests under Ar.

B7. 1. 125, please replace substantially by actual value

- Has been replaced by actual value. (Line 146 in the manuscript)

As shown in Fig. 2a, introducing TEMPO into the system decreased the conversion of active carbon from 17% to 8%.

Reviewer #2 (Remarks to the Author):

The hydrogenation of carbon-based solids into hydrocarbons is widely used in industrial applications, but it is a slow reaction since the strong carbon-carbon bonds in the carbon frameworks. This paper demonstrates this reaction can be proceeded efficiently at room temperature by mechanochemistry. The authors systematically investigated the broad range of different carbon substrates and the influence of different metal catalysts. Importantly, they took a deeper insight into the mechanism of mechanochemical hydrogenation. Using the radical scavenger, TEMPO, the proposed radical mechanism of mechanochemical hydrogenation is reliable and impressive, it could guide more exploration to catalytical reaction by novel approach. Overall, this manuscript will attract broad interest, and thus it is worthy to consider in Nat. Commun. after addressing following issues.

1. In the Extended Data Tables 2 and 3, please specify the C1-C4 products to clarify whether olefin or alkyne exists in the C2-C4 products.

- The C2-C4 are all alkanes, and this has been clarified in the main text (Line 85-86 in the manuscript) and tables.

C₂-C₄, only as alkane compounds, have also been detected.

Extended Data Table 1 | The selectivity to C₁-C₄ alkane for 7-hour milling.

Extended Data Table 2 | The selectivity to C₁-C₄ alkane for 21-hour milling.

2. For the products, the authors do not mention whether H₂ completely reacts. If there is unconsumed H₂ in the container after reaction, it should be considered in calculating the selectivity of products.

- Since the important substrate - and also the limiting substrate - to convert is the carbon-containing substrate, we refer selectivities to the carbon products. Since hydrogen is in excess, quoting selectivities for hydrogen would be not very useful. One could calculate selectivities for hydrogen in case of the oxygen containing substrates, i.e. hydrogen used for producing hydrocarbons versus hydrogen used for water formation. However, due to the difficulties in determining the small amounts of water, this is not possible.

3. The supporting material plays an important role in the catalytic reaction, and therefore it is necessary to add contrast experiments based on different supporting materials such as CeO₂, MgO and SiO₂.

- Following the suggestion of the reviewer, we have included additional supports. MgO was found to be a highly suitable support among the supports screened. After 7 hours milling, 83% conversion of active carbon was reached over MgO. In contrast, the conversions over the other supports were all below 20%. Support effects are highly diverse in heterogeneous catalysis¹⁹. To obtain a full understanding of the support effects in this novel catalytic system, a more extensive study would be necessary, which is beyond the scope of this paper. We feel that the description of the depolymerization/hydrogenation of carbon substrates is valuable as it stands, including the findings for the different supports, without being able at this point to explain the effects in detail.

The changes in the manuscript are combined for comments 3., 4., and 5. Please see section highlighted in yellow after 5.

4. The authors display the high efficiency of the first round of mechanochemical hydrogenation, please add the catalytic stability tests of Fe/ γ -Al₂O₃ catalyst.

- Following the advice of this reviewer, we found MgO was a much better support, and also that lower pressure did not affect conversion significantly. We thus tested the stability of Fe/MgO under a pressure of 20 bar. The milling time for each round was 6 hours.

In the first round, 50 mg active carbon was added. In each subsequent round, an equal amount of activated carbon was added to replenish the quantity of converted activated carbon from the previous round, ensuring a consistent initial carbon amount throughout each round. In the fifth round the conversion dropped to 44%.

The changes in the manuscript are combined for comments 3., 4., and 5. Please see section highlighted in yellow after 5.

5. The reviewer notices the pressure of H₂ is 170 bar, this is harsh condition and one thinks this experimental condition could not be considered as a mild condition which mentioned in the manuscript. Also, the small amounts of carbon-based solids and extremely high pressure may not be substantial technical leap for scale-up mechanochemical hydrogenation of carbon-based solids. Thus, the reviewer suggests checking whether this reaction can be proceeded at a lower pressure of H₂ as well as bigger scale of reaction to still obtain high efficiency by mechanochemistry.

- Following the reviewer' s suggestion, different conditions were explored. The reaction proceeds at lower pressure without significantly reduced conversion down to about 20 bar. Further decreasing the hydrogen pressure

to 5 bar, the conversion dropped to 59%. For a bigger scale experiments, we would need to build a bigger reactor and order a larger mill. However we notice that in a study of mechanochemical hydrogasification of charcoal²⁰, the scale-up reaction displayed a similar conversion to that of the small-scale

Changes in the manuscript corresponding to 3.,4. and 5. are given below (Line 247-269 in the manuscript)

In order to further explore the potential of this reaction, we tested different supports and hydrogenation/depolymerization under lower pressure. MgO was found to be a highly suitable support among the supports screened (Fig. 5a). After 7 hours of milling, 83% conversion of AC can be reached over MgO. In contrast, the conversions over the other supports were all below 20%. Support effects are diverse in heterogeneous catalysis¹⁹. To fully understand the support effects in this reaction, much more extensive studies are required which is beyond the scope of this work. However, these initial data suggest that there is substantial upside potential for these systems.

Different hydrogen pressures were compared by using MgO as

support (Fig. 2b). The conversion did not change significantly when the pressure was reduced from 170 bar to 20 bar. Upon further decreasing the hydrogen pressure to 5 bar, the conversion dropped to 59% (under these conditions, 5 bar hydrogen pressure would be sufficient to just convert the full amount of carbon present).

Under a hydrogen pressure of 20 bar, the stability of the Fe-MgO catalyst was tested. In the first round 50 mg of active carbon was added. In each subsequent round, an equal amount of activated carbon was newly added to replenish the converted activated carbon from the previous round, ensuring a consistent initial carbon amount throughout each round. The conversion dropped to 44% in the fifth round. This could be due to different factors: the surface of the catalyst may be poisoned by impurities or side products of the reactions, alternatively, the easy-to-depolymerize carbon may react first, so that after each repetition upon replenishing only the converted carbon, the difficult-to-polymerize carbon remains in the system. A detailed study of the deactivation is subject of current investigations.

Reviewer #3 (Remarks to the Author):

The manuscript describes a mechanocatalytic process for converting different carbon-based solids methane and small amounts of higher hydrocarbons. The

manuscript is quite interesting and should be published once the following issues are addressed:

1. It would be nice if the authors could discuss why PE is significantly less reactive than any of the other materials studied.

- Physically, PE is rather resistant to abrasion and impacts. In addition, it is relatively soft and plastically deformed, which leads to better dissipation of mechanical energy in the materials. It is also chemically rather stable, compared to the polymers with oxygen in the backbone, since it contains only C-C bonds. Both factors make PE less reactive²¹. However, we have explored it to a substantial extent, because it is one of the most interesting substrates, due to its high production amounts and the lack of alternatives for its depolymerization.

A discussion was added in the main text (Line 81-85 in the manuscript)

The low conversion of PE can be contributed to a combination of its physical resistance to abrasion and impacts, its relative softness and tendency for plastic deformation, dissipating mechanical energy more efficiently than the other substrates, and chemical inertness of the monotonous stable C-C bonds²¹

2. If butane conversion after 7 h of milling reaching only 10% the high methane yield from the solid feedstocks is not entirely explained by formation of intermediates and their subsequent methanation.

- The solid reaction can be much more favorable since the solid is agitated directly by the mechanical force to generate radicals. In our TEMPO trapping experiments under Ar with PE as substrate, we also observed the highest signal of C₁-TEMPO among the signal of C₁-C₆-TEMPO. This indicates that most of the methane is directly formed from CH₃• radicals rather than from C₂-C₄ intermediates.

3. In lines 110 to 114, the authors refer to earlier attempts to produce light hydrocarbons from solid feedstocks in vague terms. They should provide specific examples and references.

- We have slightly modified the text and give four examples to be more specific, as follows: (Line 129-133 in the manuscript)

The fact that essentially any carbon-containing material seems to be hydrogenated to light hydrocarbons under ball milling is highly

significant, both for its practical implications and for fundamental reasons, since such broad reactivity in hydrogenations at fairly mild conditions for solid substrates, compared to earlier attempts, which focus on one class of substrates, often under harsh conditions²²⁻²⁵, had never been observed.

4. Ruthenium has shown considerable activation for hydrothermal methanation of biomass. Thus, it would be interesting to include it in the comparison of metal catalysts.

- Following the suggestion of the reviewer, we have added Ru to the metals studied. The conversion is 15 %, close to Fe, Co and Ni. Ru has similar ability to adsorb C_2 species as Fe, Co and Ni⁹. It also produces C_2^+ products in the Fischer-Tropsch reaction. This is in line with the proposed mechanism. The adsorption of polymers is necessary for radical generation. The section discussing the different metals and their role has accordingly been supplemented by ruthenium (Fig. 3 and lines 184, 185, 208, 214 and 215).

Minor issues:

Line 81: methane should not be capitalized

Line 194: Typo in carbon-based polymers

These points have been corrected.

- 1 Kubota, K., Pang, Y., Miura, A. & Ito, H. Redox reactions of small organic molecules using ball milling and piezoelectric materials. *Science* **366**, 1500–1504 (2019).
- 2 Cheng, H., Hernández, J. G. & Bolm, C. Mechanochemical Cobalt - Catalyzed C–

- H Bond Functionalizations by Ball Milling. *Advanced Synthesis Catalysis Communications* **360**, 1800–1804 (2018).
- 3 Schmidt, R., Martin Scholze, H. & Stolle, A. Temperature progression in a mixer ball mill. *International Journal of Industrial Chemistry* **7**, 181–186 (2016).
- 4 Takacs, L. & McHenry, J. Temperature of the milling balls in shaker and planetary mills. *J Mater Sci* **41**, 5246–5249 (2006).
- 5 Singh, B. & Sharma, N. Mechanistic implications of plastic degradation. *Polymer Degradation and Stability* **93**, 561–584 (2008).
- 6 Chen, L., Qi, X., Tang, J., Xin, H. & Liang, Z. Reaction pathways and cyclic chain model of free radicals during coal spontaneous combustion. *Fuel* **293**, 120436 (2021).
- 7 Fan, Y. *et al.* A new perspective on polyethylene-promoted lignin pyrolysis with mass transfer and radical explanation. *Green Energy Environment* **7**, 1318–1326 (2022).
- 8 Tricker, A. W., Samaras, G., Hebisch, K. L., Realff, M. J. & Sievers, C. Hot spot generation, reactivity, and decay in mechanochemical reactors. *Chemical Engineering Journal* **382** (2020).
- 9 Jones, G., Studt, F., Abild-Pedersen, F., Nørskov, J. K. & Bligaard, T. Scaling relationships for adsorption energies of C2 hydrocarbons on transition metal surfaces. *Chemical Engineering Science* **66**, 6318–6323 (2011).
- 10 Han, G. F. *et al.* Mechanochemistry for ammonia synthesis under mild conditions. *Nat Nanotechnol*, 1–6 (2020).
- 11 Tricker, A. W. *et al.* Mechanocatalytic Ammonia Synthesis over TiN in Transient Microenvironments. *ACS Energy Letters* **5**, 3362–3367 (2020).
- 12 Reichle, S., Felderhoff, M. & Schuth, F. Mechanocatalytic Room-Temperature Synthesis of Ammonia from Its Elements Down to Atmospheric Pressure. *Angew Chem Int Ed Engl* **60**, 26385–26389 (2021).
- 13 Tricker, A. W. *et al.* Mechanocatalytic hydrogenolysis of benzyl phenyl ether over supported nickel catalysts. *RSC Sustainability* **1**, 346–356 (2023).
- 14 Nguyen, V. S., Chang, Y., Phillips, E. V., DeWitt, J. A. & Sievers, C. Mechanocatalytic Oxidative Cracking of Poly (ethylene) Via a Heterogeneous Fenton Process. *ACS Sustainable Chemistry Engineering* (2023).
- 15 Štrukil, V. Highly Efficient Solid-State Hydrolysis of Waste Polyethylene Terephthalate by Mechanochemical Milling and Vapor-Assisted Aging. *ChemSusChem* **14**, 330–338 (2021).
- 16 Tricker, A. W. *et al.* Stages and Kinetics of Mechanochemical Depolymerization of Poly (ethylene terephthalate) with Sodium Hydroxide. *ACS Sustainable Chemistry Engineering* **10**, 11338–11347 (2022).
- 17 Zhou, J., Hsu, T.-G. & Wang, J. Mechanochemical Degradation and Recycling of Synthetic Polymers. *Angewandte Chemie International Edition*, e202300768 (2023).
- 18 Peña-Alvarez, M. *et al.* In-situ abiogenic methane synthesis from diamond and graphite under geologically relevant conditions. *Nature Communications* **12**,

- 6387 (2021).
- 19 van Deelen, T. W., Hernández Mejía, C. & de Jong, K. P. Control of metal-support interactions in heterogeneous catalysts to enhance activity and selectivity. *Nature Catalysis* **2**, 955–970 (2019).
- 20 Han, G. F. *et al.* Extreme Enhancement of Carbon Hydrogasification via Mechanochemistry. *Angew Chem Int Ed Engl* **61**, e202117851.
- 21 Vasile, C. & Pascu, M. *Practical guide to polyethylene*. (iSmithers Rapra Publishing, 2005).
- 22 Yasuda, H., Yamada, O., Zhang, A., Nakano, K. & Kaiho, M. Hydrogasification of coal and polyethylene mixture. *Fuel* **83**, 2251–2254 (2004).
- 23 Tomeczek, J. & Gil, S. The kinetics of coal chars hydrogasification. *Fuel Processing Technology* **91**, 1564–1568 (2010).
- 24 Zhou, H., Wang, M. & Wang, F. Oxygen-vacancy-mediated catalytic methanation of lignocellulose at temperatures below 200° C. *Joule* **5**, 3031–3044 (2021).
- 25 Ji, H. *et al.* Boosting Polyethylene Hydrogenolysis Performance of Ru - CeO₂ Catalysts by Finely Regulating the Ru Sizes. *Small*, 2300903 (2023).

REVIEWERS' COMMENTS

Reviewer #1 (Remarks to the Author):

I have reviewed the corrections made by the authors and believe they have addressed them all in the best of their ability. I recommend acceptance

Reviewer #2 (Remarks to the Author):

This ready for publication now.

Reviewer #3 (Remarks to the Author):

The authors addressed all my comments. I recommend acceptance of the manuscript in its present form.